# Exploring Matrix Effects on Binding Properties and Characterization of Cotinine Molecularly Imprinted Polymer on Paper-Based Scaffold

**DOI:** 10.3390/polym11030570

**Published:** 2019-03-26

**Authors:** Nutcha Larpant, Yaneenart Suwanwong, Somchai Boonpangrak, Wanida Laiwattanapaisal

**Affiliations:** 1Graduate Program in Clinical Biochemistry and Molecular Medicine, Faculty of Allied Health Sciences, Chulalongkorn University, Bangkok 10330, Thailand; Nutchalp.l@gmail.com; 2Department of Clinical Microscopy, Faculty of Allied Health Sciences, Chulalongkorn University, Bangkok 10330, Thailand; Yaneenart.S@chula.ac.th; 3Center for Research and Innovation, Faculty of Medical Technology, Mahidol University, Nakhon Pathom 73170, Thailand; somchai.boo@mahidol.ac.th; 4Department of Clinical Chemistry, Faculty of Allied Health Sciences, Chulalongkorn University, Bangkok 10330, Thailand; 5Electrochemistry and Optical Spectroscopy Center of Excellence, Chulalongkorn University, Bangkok 10330, Thailand

**Keywords:** molecularly imprinted polymer, adsorption capacity, paper-based scaffold, cotinine

## Abstract

Commercially available sorbent materials for solid-phase extraction are widely used in analytical laboratories. However, non-selective binding is a major obstacle for sample analysis. To overcome this problem, molecularly imprinted polymers (MIPs) were used as selective adsorbent materials prior to determining target analysts. In this study, the use of non-covalent molecularly imprinted polymers (MIPs) for cotinine adsorption on a paper-based scaffold was studied. Fiberglass paper was used as a paper scaffold for cotinine-selective MIP adsorption with the use of 0.5% agarose gel. The effects of salt, pH, sample matrix, and solvent on the cotinine adsorption and extraction process were investigated. Under optimal conditions, the adsorption isotherm of synthesized MIPs increased to 125.41 µg/g, whereas the maximum adsorption isotherm of non-imprinted polymers (NIPs) was stable at 42.86 µg/g. The ability of the MIP paper scaffold to absorb cotinine in water medium was approximately 1.8–2.8-fold higher than that of the NIP scaffold. From Scatchard analysis, two dissociation constants of MIPs were calculated to be 2.56 and 27.03 µM. Nicotine, myosmine, and N-nitrosonornicotine were used for selectivity testing, and the calculated selectivity factor of cotinine to nicotine, myosmine, and N-nitrosonornicotine was 1.56, 2.69, and 2.05, respectively. Overall, the MIP paper scaffold is promising for simple onsite sampling of cotinine and can be used to assess tobacco smoke exposure.

## 1. Introduction

Tobacco smoking is one of the risk factors associated with non-communicable diseases. Additionally, environmental tobacco smoke is one of the causes of air pollution, which has a great impact on human health, especially in children and pregnant women [1]. For many medications, cigarette smoke can induce alterations of pharmacokinetics and drug metabolism, resulting in inappropriate dosage adjustment [2,3], and in some cases of diagnosis, tobacco smoke can also interfere with results [4]. Thus smoking is not allowed before investigating some substances (e.g., blood ammonia). However, self-reporting of tobacco use is not reliable, as approximately 35% of smokers do not confess their smoking, according to a report on patients before surgery [5]. Interestingly, there is evidence from meta-analyses and systematic reviews for an increased risk of postoperative complications, including morbidity, general infections, pulmonary complications, neurological complications, wound complications, and admission to the intensive care unit, associated with preoperative smoking [6]. Apart from interviews and history-taking with patients, an appropriate way to obtain reliable information about whether they have used cigarettes is needed. Detection of the metabolites and chemical constituents of tobacco exposure, including nicotine, benzopyrene, and thiocyanate, from body fluid specimens, e.g., urine, plasma, and saliva, is the target of smoking assessment. Among those biomarkers, cotinine, a major metabolite of nicotine, is widely recommended for monitoring of tobacco smoke exposure because of its long half-life and distribution in various bodily fluids including blood, saliva, and urine [7].

There are various methods of cotinine detection, including lateral flow immunochromatography [8,9,10], radioimmunoassay [11,12], enzyme linked immunosorbent assay [13,14,15], and colorimetric determination [16,17,18]. Immunochromatography is one of the most popular rapid testing platforms for preliminary screening of cotinine, since it is simple and easy to use. However, variables of individual color perception among users and semiquantitative methods have been addressed as limitations of this naked-eye detection [19]. For some patients, highly sensitive methods such as gas chromatography (GC) [20,21], high-performance liquid chromatography (HPLC) [22,23], and liquid chromatography–mass spectrometry (LC-MS) [24,25] are still in demand for accurate quantitation of cotinine to provide the exact status of nicotine exposure. For all of these techniques, sample preparation is a crucial step in the analytical method because most real samples cannot be introduced to those instruments directly, due to many factors that can affect analytical results depending on the sample matrix, type of analytical technique, and range of concentration level [26]. Sample preparation consisting of adsorption and extraction of tobacco alkaloids in human specimens to an adsorbent material prior to determination has been developed and studied [27].

The extraction of targets of analysis from biological samples is challenging. Although classical approaches for sample preparation using solid- or liquid-phase extraction can adsorb the target analyte, other chemicals are retained in the eluting sample [28]. Commercially available sorbent materials for solid-phase extraction are widely used in most analytical laboratories. However, non-selective binding is a major problem for sample analysis.

Molecularly imprinted polymers (MIPs) are now attracting much interest in a number of applications, including sample pretreatment [29], chromatographic separation, and chemical or biosensing elements. In particular, the molecular imprinting technique provides synthetic materials composed of specific binding sites that recognize target analytes. Typical components in the process of MIP production are templates, monomers, crosslinkers, porogens, and initiators. Besides the traditional production processes of MIP, advanced MIP strategies have been developed, including multifunctional monomer imprinting, dummy imprinting, segment imprinting, composite material imprinting, and multitemplate imprinting strategies [30]. Moreover, this technique is considered versatile, robust, reproducible, and cost-effective compared with the antibody-based method [31,32,33]. MIPs can cooperate with other sensing elements, including surface plasmon resonance or infrared spectroscopy sensing [34], colorimetric or ultraviolet (UV)-visible sensing [35], surface-enhanced Raman scattering sensing [36,37], chemiluminescence sensing [38], electrochemical sensing [39], paper spray ionization mass spectrometry [40], and fluorescence sensors [41]. Analytes in samples should be enriched or preconcentrated to an optimal concentration prior to analysis, thus sample preparation plays an important role in quantitative determination. Recently, there have been many reports on composite materials as supporting substrates for MIPs, including filter paper and glass fiber membrane [42]. Many platforms have been designed for various application strategies, such as determination of organic compounds [43], human carcinogens [44], ocean pollutants [45], pesticides [46,47], and 17β-estradiol [48]. The benefits of paper as a component material in a variety of areas include high porosity, degradability, no need for external power supply, and low cost [49]. These advantages have encouraged the use of paper in biosensor development. Recent advances in the fabrication and modification of paper-based platforms have led to the feasible use of paper in a wide range of sample types and sample preparations [50].

Glass fiber membranes have been widely used for sample preparation in processes such as DNA extraction [51] and removal of particles or sediments in air sampling [52]. A glass fiber membrane has also been used as a supporting substrate that has superior ability to enrich trace metals before analysis [53]. Interestingly, there have been reports on the use of glass fiber membrane incorporated with MIPs for determination of phenolic compounds [54]. However, no studies on kinetic adsorption of target analytes on MIP glass fiber membrane–based scaffold with agarose gel assistance have been reported.

In the present work, an MIP paper-based scaffold is introduced. Glass fiber membranes and agarose gels were implemented for the immobilization of MIP particles. Glass fiber membranes are commonly used as the conjugated pad in immunochromatography, which is able to release labeled conjugate and low-protein binding [55]. The advantages allow researchers to apply glass fiber membranes as part of paper scaffolds. Moreover, agarose gel, a common reagent used in many molecular laboratories, was applied for the temporary immobilization of cotinine MIPs on the glass fiber membrane. The effects of salt, pH, sample matrix, and solvent on cotinine adsorption and extraction were investigated for the production of suitable MIPs. Based on our study, MIP paper-based scaffold could potentially be used for sample collection in rural areas before being tested in a central laboratory due to its advantages of convenient use, easy preparation, portability, and stability in a wide temperature range.

## 2. Materials and Methods

### 2.1. Chemicals and Reagents

Cotinine, nicotine, myosmine, N-nitrosonornicotinine, methacrylic acid (MAA), ethylene glycol dimethacrylate (EGDMA), azobisisobutyronitrile (AIBN), potassium phosphate monobasic (KH_2_PO_4_), sodium phosphate dibasic (Na_2_HPO_4_), and agarose powder were purchased from Sigma-Aldrich (St. Louis, MO, USA). MAS^®^ DOA TOTAL Control liquid assayed drugs of abuse control was purchased from Thermo Fisher Scientific (Waltham, MA, USA). High-performance liquid chromatography (HPLC) grade chemical solvents including acetonitrile, dichloromethane, acetic acid, 2-propanol, and methanol were purchased from Sigma-Aldrich and Merck (Kenilworth, NJ, USA). Ammonia (30%) was purchased from Panreac Quimica SAU (Barcelona, Spain). Glass fiber (GF) membrane filter was purchased from a local manufacturer in Thailand (Pacific Biotech CO. LTD, Phetchabun, Thailand). Filter paper (Whatman, Grade 1) was purchased from GE Healthcare Life Science (Chicago, IL, USA). CHROMABOND^®^ column C_18ec_ was purchased from Macherey-Nagel GmbH & Co. KG (Düren, Germany). Deionized water was purified by a Milli-Q system (Merck Millipore, Billerica, MA, USA) with resistivity of 18.2 Ω cm.

### 2.2. Preparation of Cotinine-Imprinted Polymers

Cotinine-imprinted polymers were prepared as previously described [56], with minor modifications. Briefly, 1 mmol cotinine, 4 mmol MAA, and 5.6 mL dichloromethane were mixed in a glass test tube (13 × 100 mm), then 20 mmol EGDMA and 0.24 mmol AIBN were added and mixed thoroughly. The prepolymerized mixture was deoxygenated by purging with nitrogen gas for 10 min. After that, polymerization was allowed to occur by incubating the mixture at 35 °C for 8 h in a water bath shaker, followed by increasing the temperature to 60 °C for 16 h. To produce the non-imprint polymer, the control polymers were formed in parallel by using the same protocol, except for the addition of cotinine templates in the reaction. Finally, rigid polymers were obtained and crushed to fine particles by mortar and pestle. To extract the template from the imprinted polymer, a mixture of methanol and acetic acid (9:1 ratio) was used, and the synthesized polymers were then washed with methanol. The washing step was repeated at 4 hour intervals for 12 cycles until the cotinine in the solution was not detected by HPLC. In the final step, each polymer was obtained by centrifugation and subsequently allowed to dry in a hot air oven at 65 °C for 24 h.

### 2.3. Preparation of Molecularly Imprinted Polymer (MIP) Paper-Based Scaffolds

Unless otherwise stated, for preparation of each batch of paper-based scaffold, 10 mg of MIP was mixed with 250 µL of 0.5% melted agarose gel by using a microcentrifuge tube. This condition was selected based on a compromise between gel volumes that evenly adsorbed to stacks of 4 papers without the agarose gel overflowing from the papers. The percentage of agarose gel used in this condition was based on time to polymerization. Here, a concentration of gel higher than 0.5% caused too rapid polymerization and resulted in uneven dispersion of MIP particles on the papers. On the other hand, agarose gel lower than 0.5% caused too slow polymerization and an inability to sustain MIP particles adhering to the surface of paper stacks. The MIP–agarose gel mixture solution was transferred to circular stacks of 4 glass fiber membranes (0.6 cm in diameter, 0.05 cm in thickness). MIP paper scaffolds were incubated at 65 °C in a hot air oven for 6 h. Finally, the MIP paper-based scaffolds were used for cotinine adsorption. A schematic of MIP paper-based scaffold preparation is shown in Figure 1. NIP paper-based scaffolds were also prepared with the same protocol. Microscopic images of glass fiber membranes observed under a stereo microscope (9.7×) and MIP paper-based scaffold are also shown in the Appendix A.

### 2.4. Polymer Characterization

#### 2.4.1. Study of Size and Shape of Synthesized Polymer

Field emission scanning electron microscopy (FESEM) was employed to investigate the microstructure and morphology of the synthesized imprinted polymers. Different modes, including lower secondary and upper secondary electron in-lens modes, were used to observe polymer morphology. In addition, gentle beam mode, which reduces charge buildup and high resolution at very low voltages, was used for surface observation. These topographical studies were carried out with a JEOL model JSM-7610F (JEOL, Tokyo, Japan). Dried synthesized polymers were placed on metallic sample holders and coated with gold by a sputter coater. Prepared samples were analyzed under FESEM at an accelerating voltage of 2 kV. Micrographs of all samples were recorde d at magnifications ranging from 50× to 30,000×.

To study the chemical interaction between components of polymers, Fourier transform–infrared spectroscopy (FT–IR) (Perkin Elmer, Spectrum One) was employed for characterization of imprinted polymers. Polymer powder (3 mg) was deposited on a diamond holder and processed by means of transmitting the infrared radiation in the range of 4000–515 cm^−1^.

#### 2.4.2. Optimization of the Cotinine Binding Experiment

To optimize and study the effect of each parameter on cotinine binding and extraction, the HPLC method was used for the whole experiment. Cotinine determination was performed using HPLC with 260 nm UV detection at 40 °C. A reverse phase column, Purospher® STAR RP-18 endcapped (5 µm) LiChroCART® 250-4.6 (250 mm × 4.6 mm id, particle size 5 µm, Merck), was used for separation. The mobile phase consisted of 71% ammonium acetate buffer (13 mM) and 29% acetonitrile. The pH of the mobile phase solvent was adjusted to 5.0 by glacial acetic acid. Samples were filtered through a 0.45 µm syringe filter before injection. The injection volume was 20 µL and the flow rate was set at 0.8 mL/min. Cotinine standards in a concentration range of 62.5–500 µg/mL with internal standard (2-phenylimidazole) were run as the mentioned condition. The chromatogram of standard mixture is shown in the Appendix A.

#### 2.4.3. Effect of pH and Salt Ions

To study the effect of pH, cotinine standard solutions in each pH from 5–8 were incubated with MIP paper scaffolds for 2 h. To study the effect of salt ions on adsorption, NaCl was added to a standard solution of cotinine for investigation of adsorption capacity and extraction. NaCl solution in the range of 1.25–10% *w*/*v* (pH 7.0) was used to study the effect on cotinine adsorption and extraction. In the extraction process, MIP and NIP paper-based scaffolds were washed with 3 mL of water and eluted with a mixture of methanol and acetic acid (9:1). The solution obtained after the adsorption and extraction process underwent determination by HPLC.

#### 2.4.4. Effect of Solvent on Binding

Solvents, including methanol and acetonitrile, were employed to investigate their effect on cotinine adsorption capacity. Cotinine was dissolved in each constituent of 25%, 50%, 75%, and 100% methanol or acetonitrile. Then MIP and NIP particles were incubated with prepared solvents for investigation of adsorption capacity. After adsorption, cotinine remaining in the supernatant was determined by HPLC-UV.

#### 2.4.5. Kinetic and Adsorption Isotherm Experiment

To investigate adsorption dynamics, an MIP paper-based scaffold was placed in a microcentrifuge tube and mixed with a standard solution of cotinine (62.5–4000 µg/mL). The concentration of cotinine remaining after adsorption for 2 h was determined by HPLC-UV at 260 nm. For the adsorption isotherm experiment, after adsorption of cotinine standard by MIP at different incubation times (10–300 min), the amount of cotinine adsorbed to each polymer was calculated using the following equation [57]:
(1)$$Q=\frac{\left(C_{0}-C_{e}\right)\times V}{W},$$
where *Q* is the adsorption capacity, *C*_0_ is the initial cotinine concentration (µg/mL), *C_e_* is the equilibrium of cotinine concentration at different time intervals (µg/mL), *V* is the volume of cotinine standard solution (mL), and *W* is the weight of the dry polymer (g).

The equilibrium dissociation constant (*K_d_*) of MIPs and NIPs was further determined using the following Scatchard equation [58]:(2)QC=(Qmax−Q)Kd,
where *Q* is the amount of cotinine bound to the polymer, *Q*_max_ is the maximum adsorption amount of cotinine on the polymer, and *C* is the equilibrium of cotinine concentration (µg/mL).

#### 2.4.6. Selectivity of the Synthesized Polymer

Testing the selectivity of the synthesized polymer was conducted. Structures similar to cotinine, including nicotine, myosmine, and N-nitrosonornicotinine, were used for the rebinding experiment. Standard solutions of nicotine, myosmine, N-nitrosonornicotinine, and cotinine were incubated with MIPs and NIPs for 2 h. The remaining amount of cotinine or cotinine analog in the supernatant was determined by HPLC. The imprinting factor (IF) can be determined by the following equation:

IF = Q MIP/Q NIP
(3)


In addition, the selectivity factor (α) can be calculated as follows:

Selectivity (α) = IF cotinine/IF cotinine analog
(4)


#### 2.4.7. Interference Study on Adsorption Capacity of MIP and Recovery Study on Extraction Process

Cotinine standard at concentrations of 62.5 and 125 µg/mL was spiked into control urine to obtain low and high cotinine levels before assay. To investigate the effect of interference, 125 µg/mL nicotine was spiked into the urine control samples. The same amounts of MIP particles and MIP-paper based scaffold were separately incubated with cotinine-spiked urine control for 2 h. Then, adsorption capacity was evaluated by HPLC.

To demonstrate the applicability of the MIP paper-based scaffolds for heavy tobacco use, cotinine standard at 5 µg/mL was spiked into matrix samples (e.g., water and urine control), in which nicotine (2.5 and 5 µg/mL) was also spiked, to test the robustness. Recovery of the cotinine in a water and urine matrix with the presence of nicotine was investigated.

#### 2.4.8. Application and Method Comparison of MIP Paper-Based Scaffold

First, conditioning of the MIP paper-based scaffold was performed by applying 200 µL of methanol and 200 µL of Milli-Q water. Then the conditioned MIP paper-based scaffold was incubated with 250 µL of urine control (3–10 µg/mL) for 2 h. After the adsorption process, the scaffold was transferred to a microcentrifuge tube containing 500 µL of dichloromethane and 500 µL of dichloromethane:2-propanol:25% ammonia (8:2:0.2). The eluted solution was evaporated to dryness under nitrogen. Finally, it was resuspended with 250 µL Milli-Q water for further HPLC analysis.

In the method comparison, solid-phase extraction columns (CHROMABOND^®^ column C_18ec_) [27] were used to extract cotinine in urine control by the following protocol with some modification: 1 mL of urine control (3–10 µg/mL) was added with 200 µL of 0.01 mol/L NaOH. The column was applied with 2 × 2 mL methanol, then 2 × 2 mL water, then the prepared sample was applied through the column. Next, the washing step was performed by adding 3 mL of distilled water. The column was dried under nitrogen. In the eluting step, 1.5 mL of dichloromethane, then 1 mL of dichloromethane:2-propanol:25% ammonia (8:2:0.2) were added. The eluted solution was evaporated to dryness at 50 °C with a gentle stream of nitrogen. Samples were resuspended in 250 µL Milli-Q water for further measurement by HPLC and the results from both methods were compared and plotted as a Band–Atman plot.

## 3. Results

### 3.1. Morphological Characteristics

FESEM was employed to observe the surface morphology of MIPs and NIPs. The morphological structure of MIPs (Figure 2a) and NIPs (Figure 2b) shows a rough porous surface and uneven shape due to the mechanical grinding process. There are no significant differences in cavity size between the MIP and NIP particles. Figure 2c shows an untreated glass fiber with numerous large cavities between fibers; thus, MIP particles can be temporarily immobilized by agarose coating. Figure 2e shows the MIP paper-based scaffold. When compared with MIPs immobilized on glass fiber, a small amount of MIPs were adsorbed onto the filter paper (Figure 2g) due to the small diameter of the pores (11 µm).

### 3.2. Attenuated Total Reflection Fourier Transform–Infrared (ATR-FT–IR) Analysis

To investigate the chemical interaction of the polymers, ATR-FT–IR was performed. The graph shows the small differences among the three lines; however, MIP (nonextract) provides the difference in % transmittance along the range 515–4000 cm^−1^ when compared to MIP (extracted) and NIP (Figure 3).

Regarding the spectra of MAA and EGDMA, significant bands at 1633 and 1636 cm^−1^ were assigned to C=C stretching [32]. Synthesized MIPs and NIPs also presented two significant peaks. Although both cotinine-MIPs and NIPs also exhibited the band corresponding to C=O stretching, the band of nonextracted MIPs was slightly shifted to the lower wave number (1719 and 1720 cm^−1^ for nonextracted MIP and NIP, respectively) (Appendix A). The band corresponding to C=O stretching in extracted MIPs was slightly larger than in the nonextracted MIPs. This demonstrated the formation of H-bonds between the functional monomer and cotinine.

### 3.3. Adsorption Isotherm and Adsorption Kinetics of Synthesized Polymers

A saturation adsorption experiment was conducted using different concentrations of a cotinine standard (Figure 4a). The remaining cotinine concentration was detected by HPLC. The linearity of HPLC method was found in range from 0.5-500 µg/mL, and the limit of detection of 0.35 ng/mL (S/N = 3) was obtained. The amount that bound to MIPs was significantly higher than the amount that bound to NIPs at all concentrations of cotinine (Student’s *t*-test; p < 0.05). The adsorption capacity of synthesized MIPs increased from 4.41 to 125.41 µg/g when the concentration of cotinine was increased from 62.5 to 4000 µg/mL. Although NIPs showed a similar pattern of cotinine-MIP adsorption, their adsorption capacity remained stable at 42.86 µg/g. This indicates that even at a high concentration of cotinine, MIPs were able to continually adsorb target analytes. The binding sites of MIPs with high affinity to cotinine were formed in large quantity during the polymerization process [59]. Kinetic adsorption of synthesized polymers was observed from 10 min to 300 min (Figure 4b). Although the adsorption rate of both MIPs and NIPs reached equilibrium within 2 h, the maximum adsorption capacity of MIPs was almost twofold higher than that of NIPs. This indicates that a large amount of cotinine can be adsorbed in a shorter period of time compared with NIPs. This rapid adsorption makes synthesized MIPs appropriate for solid-phase extraction [60]. Especially, the adsorption capacity of NIP and MIP paper-based scaffold (Figure 4b) was higher than NIP and MIP particles alone, because the paper-based scaffold delayed the release of target analyte. Moreover, the paper-based scaffolds without MIP particles could non-specifically adsorbed cotinine six times lower than MIP paper-based scaffold (Appendix A). Agarose is a naturally derived polysaccharide polymer capable of controllable fluid transport through tunable pore size and porosity [61,62]. This biopolymer has many general advantages, including hydrophilic permeability, compatibility with various types of buffers, stability under a wide range of pH, and excellent mechanical properties [63]. Based on its benefits, agarose was used as immobilizing gel material for application of functional layer by layer capsules [64] and preparation of optical sensor [65]. Furthermore, this biomaterial incorporated with other paper-based materials was implemented to control flow rate in lateral flow immunochromatography in order to enhance the sensitivity of the reaction [66]. As a result, the use of agarose gel as immobilizing medium for layered MIP on paper-based scaffold can enhance the adsorption capacity when compared with cotinine-MIP particles alone.

### 3.4. Scatchard Analysis

To estimate the equilibrium dissociation constant of MIPs and NIPs, the Scatchard equation was used as an explanation model. The Scatchard analysis provides advantages for binding characteristics of synthesized polymers, and this model is commonly used in the determination of solid-material heterogeneity, such as MIPs and NIPs. As shown in Figure 5, there are two distinct Kd values, which were obtained from the relationship between Q/C0 and Q. The equations of the left and right linear regressions of MIPs (Figure 5) were Q/C = −0.391Q + 14.243 (r^2^ = 0.99) and Q/C = −0.037Q + 9.332 (r^2^ = 0.91), respectively. The first dissociation constant (Kd1) was determined to be 2.56 µM, which indicates low affinity with a large amount of unoccupied binding sites. The second dissociation constant (Kd2) was determined to be 27.03 µM, which indicates higher affinity of MIPs with target molecules, with a small amount of unoccupied binding sites. It could be concluded that these two dissociation constants of cotinine-MIPs are a consequence of nonhomogeneity of polymer binding sites due to noncovalent interaction in the polymer matrix. During polymerization, two types of binding sites are formed: the first is a complete interaction between the functional monomer and the template, which is supposed to be a specific binding site of the MIP, and the second is a free functional monomer in the polymer matrix that causes non-specific binding [67]. In contrast to MIPs, a Scatchard plot derived from NIPs (Appendix A) providing a small correlation derived from the NIP equation indicates that cotinine can be adsorbed on the surface of control polymers by nonselective interaction, including van der Waals force, although NIPs do not contain imprinting sites [68].

### 3.5. Effect of NaCl and pH on Cotinine Adsorption and Extraction

Many chemical and biological phenomena are affected by ionic strength. Interactions between template molecules and MIPs depend on hydrogen bonding [69]. Salt ions are commonly added during the extraction process to decrease interference from water molecules in the MIP binding sites [70]. Figure 6a shows the effect of NaCl on the adsorption capacity of MIP/NIP paper scaffolds. MIPs can adsorb nearly twofold greater cotinine than NIPs at 1.25% *w*/*v* NaCl, indicating that this concentration provides the maximum absorbed cotinine ratio (QNIP/QMIP). The adsorbed cotinine ratio between MIPs and NIPs was reduced when the concentration of NaCl was greater than 1.25% *w*/*v*. In accordance with the results obtained from the cotinine adsorption study, the maximum yield of cotinine extraction was also obtained with 1.25% *w*/*v* NaCl (Figure 6b). To enhance the maximum cotinine adsorption on MIPs, 1.25% *w*/*v* NaCl was implemented for extraction. The optimal concentration of NaCl can reduce weak interactions or nonspecific binding, whereas higher concentrations of NaCl interfere with the affinity of specific binding of absorbent to target analytes [71].

To study the effect of pH on cotinine adsorption and extraction, the stock cotinine standard solution was diluted in phosphate buffer to obtain a final pH in the range of 5–8, which is the physiological range of human urine. Results demonstrated that pH 8.0 provides the maximum percentage of imprint factor for cotinine adsorption (Figure 6c), in which the cotinine was also extracted at the highest ratio (Figure 6d). This implied that using pH 8.0 as an adsorption buffer is applicable and suitable for both cotinine adsorption and further extraction. According to many reports, alkaline conditions for cotinine extraction can provide optimum recovery and a low percentage of relative error [72,73]. The pKa of cotinine is 4.5, thus a high pH reflects an optimal condition in which cotinine is in unionized form and is easily extracted from adsorbents [74].

### 3.6. Effect of Sample Matrix and Solvent on Cotinine Adsorption on MIP Paper-Based Scaffolds

Urine control was used for simulation of real sample analysis. MIP paper-based scaffolds were used for cotinine adsorption and extraction at different concentrations. Figure 6e shows that the imprinting effect ratios varied from 1.7 to 2.8. The maximum peak area ratio obtained from cotinine extraction (Figure 6f) was obtained at 2.2 µM of cotinine, which was in good agreement with the results obtained from the adsorption study. Results from the effect of sample matrix on cotinine adsorption provided a similar adsorbed cotinine ratio, which was determined using a cotinine standard solution in the rebinding experiment. These results indicated that MIP paper-based scaffolds could potentially be used for the adsorption of cotinine in a urine matrix. Inconsistent results derived from the adsorption and extraction experiments may be due to unspecific adsorption of a component in the urine control matrix. According to Caro et al., the cleanup process of the urine sample matrix following absorption by MIPs can potentially increase the extraction efficacy of target analytes [75]. To apply MIP paper-based scaffolds in aqueous solutions, different content mixtures of aqueous methanol and acetonitrile were implemented. The results showed that the highest peak area ratio of MIPs and NIPs was obtained with the use of 75% methanol and acetonitrile in the adsorption experiment, whereas the imprinting factor obtained with water medium still provided an imprinting effect of 1.8. This suggested that MIP paper-based scaffolds could be feasible for onsite urine sampling. This result correlated with another study reporting that optimized MIPs can be potentially used in aqueous environments [76,77].

### 3.7. Interference Study and Selectivity of Synthesized Polymers

From an interference study of MIP and NIP in control urine matrix (Figure 7), the adsorption capacity of MIP to 125 and 250 µg/mL of cotinine compared to cotinine-spiked nicotine decreased by 11.3% and 22.4%, respectively. In contrast to NIP, adsorption capacity increased 65.76% and 52.32% at 125 and 250 µg/mL cotinine concentration, respectively. The difference in adsorption capacity of MIP paper-based scaffolds indicates the robustness of nicotine as interference in the urine control sample matrix. Additionally, recovery of the extraction process was tested. Table 1 shows recovery of MIP and NIP paper-based scaffolds in urine matrix and water with the presence of nicotine at ratios of 2:1 and 1:1. Recovery of cotinine extraction at ratios of cotinine and nicotine of 2:1 and 1:1 was 87% and 68% in control urine and a 1% decrease in extracted yield in water, whereas recovery of NIP (urine control) increased from 72% to 77% and there was a 7% increase in extracted yield (water), which corresponded to the decreased adsorption capacity of MIP and the increased capacity of NIP with the presence of nicotine (Figure 7). The results indicated that MIP paper-based scaffold can be used in the presence of interference (nicotine) with a <50% constituent. Selectivity of MIPs was investigated with cotinine analogs (nicotine, myosmine, N-nitrosonornicotinine, and cotinine) as shown in Figure 8. The imprinting factors of MIPs toward nicotine, myosmine, N-nitrosonornicotinine, and cotinine are 2.13, 1.23, 1.63, and 3.34, respectively. Regarding the resemblance of analogs to cotinine (176.22 g/mol), the selectivity factors of cotinine to nicotine, N-nitrosonornicotinine, and myosmine were determined to be 1.56, 2.05, and 2.69, respectively. As the calculated selectivity factor, nicotine can interact with the binding site of cotinine-MIPs, providing the lowest selectivity due to the lower molar mass of nicotine (162.23 g/mol) and its smaller size when compared with cotinine. Notably, only the carbonyl group contained in the cotinine structure provided a difference from the nicotine structure. Cotinine and its analogs contain the same pyridine ring structure [20]. For example, N-nitrosonornicotinine, a potent carcinogen in an animal model [78], with molar mass closest to cotinine (177.20 g/mol) compared to other analogs, can also bind to the pocket of MIP resulting from hydrogen bonding between the nitrosonium group in N-nitrosonornicotinine and MIP matrix. This is another genotoxic alkaloid found in tobacco leaves, and its structure is related to nicotine. This substance has the smallest size and lowest molar mass (146.19 g/mol) among the three cotinine analogs, resulting in the highest selectivity factor. From the results and discussion, it can be suggested that except for their resemblance in size and molar mass, which affect MIP binding capacity, the oxygen atom in cotinine and its analog structure plays a significant role in the binding affinity of MIPs.

### 3.8. Application and Comparison of MIP Paper-Based Scaffold

Cotinine-spiked urine control in the range of 3–10 µg/mL was used to mimic real samples (n = 6) and were assayed with the proposed MIP paper-based scaffold. The results were compared with a commercial solid-phase extraction (SPE) method and demonstrated that no bias for determination of cotinine of any samples was obtained, because the difference between the two methods fell within the mean ±1.96 standard deviation (SD), as shown in Figure 9. A chromatogram after extraction by MIP paper-based scaffold is shown in Appendix A. The proposed method was not able to distinguish between MIP and NIP for cotinine levels lower than 1 µg/mL, so it is not applicable for low tobacco exposure or secondhand smoke. However, the sensitivity of the method can be increased by increasing the number of stacks of MIP paper scaffolds. Nevertheless, the proposed MIP paper-based scaffold is still merit to apply with regular active smokers that usually have cotinine concentration in urine about 1–8 µg/mL [79].

## 4. Conclusions

In this study, MIP paper-based scaffolds were successfully developed and characterized, simple and inexpensive facilities were applied for their preparation. Glass fiber membrane/cotinine-MIP hybrid materials were used for investigation of adsorption capacity, while agarose gel was used as a biologically derived immobilizing agent for layering MIP particles on the paper-based scaffold. The benefits of this layering method are the ease of fabrication and the lack of surface treatment process. In contrast to NIP, MIP paper-based scaffold provides the highest adsorption capacity. As regards the merits of exploiting agarose gel in cooperation with glass fiber membrane filter for making MIP paper-based scaffolds, this can increase the permeability of fluid through the heterogeneous pores of agarose gel, which can enhance the adsorption capacity compared with MIP particles alone. Apparently, the synthesized MIPs exhibited an affinity for cotinine template molecules compared with control polymers. From the interference and selectivity testing, the results show robustness against cotinine analog. The recovery of the extraction process in urine control and water matrix with the presence of nicotine also showed the capability of using MIP paper-based scaffolds, which offers potential use in the presence of interference (nicotine) lower than 50% constituent. In addition, the selectivity of MIPs toward cotinine was higher than nicotine, myosmine, and N-nitrosonornicotinine, indicating the specific binding of MIPs to cotinine target molecules. Interestingly, the imprinting effect obtained from the rebinding experiment in an aqueous environment still provided good results, which can potentially be used in biological samples. A comparison of methods between commercial SPE column and the proposed MIP paper-based scaffold indicated potential practical use. In summary, MIP paper-based scaffolds can be feasible for real sample preparation. Immobilization of MIP particles on paper provides a ready-to-use platform for onsite sample preparation with specific binding to cotinine prior to performing other processes in a central laboratory.

## Figures and Tables

**Figure 1 polymers-11-00570-f001:**
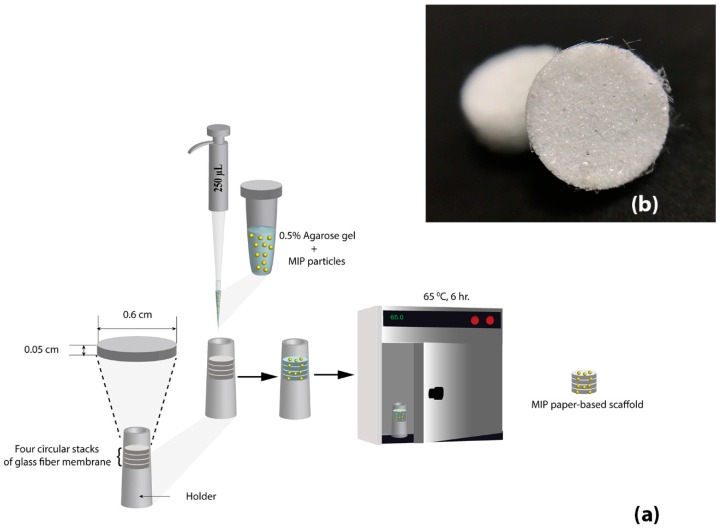
(**a**) Schematic of molecularly imprinted polymer (MIP) paper-based scaffold preparation. (**b**) Microscopic image of MIP paper-based scaffold observed under a stereo microscope (9.7×).

**Figure 2 polymers-11-00570-f002:**
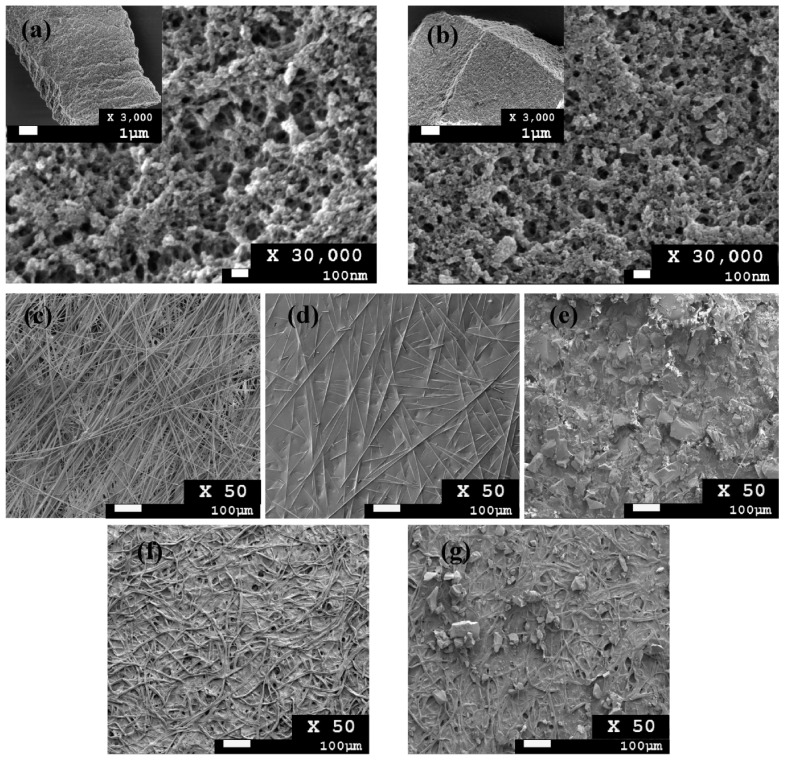
Structural morphology of (**a**) non-imprinted polymers (NIPs), (**b**) MIPs, (**c**) glass fiber membrane, (**d**) glass fiber membrane with 0.5% agarose gel, (**e**) MIP paper scaffold, (**f**) filter paper (Whatman No. 1), and (**g**) MIPs immobilized on filter paper.

**Figure 3 polymers-11-00570-f003:**
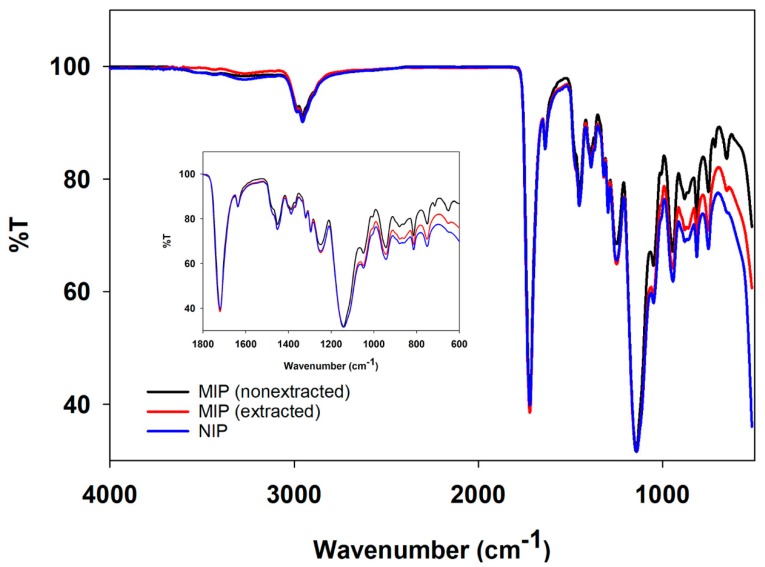
Infrared spectra of synthesized polymers. Red line represents MIP (non-extracted), blue line represents MIP (extracted), and black line represents NIP.

**Figure 4 polymers-11-00570-f004:**
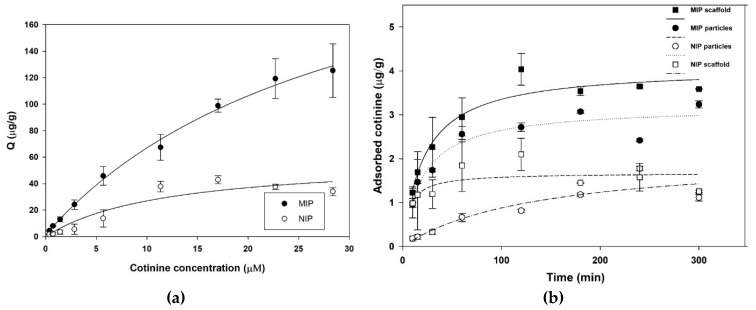
(**a**) Adsorption isotherm curve of MIPs and NIPs. (**b**) Kinetic adsorption curve of synthesized MIPs, NIPs, MIP paper scaffold, and NIP paper scaffold.

**Figure 5 polymers-11-00570-f005:**
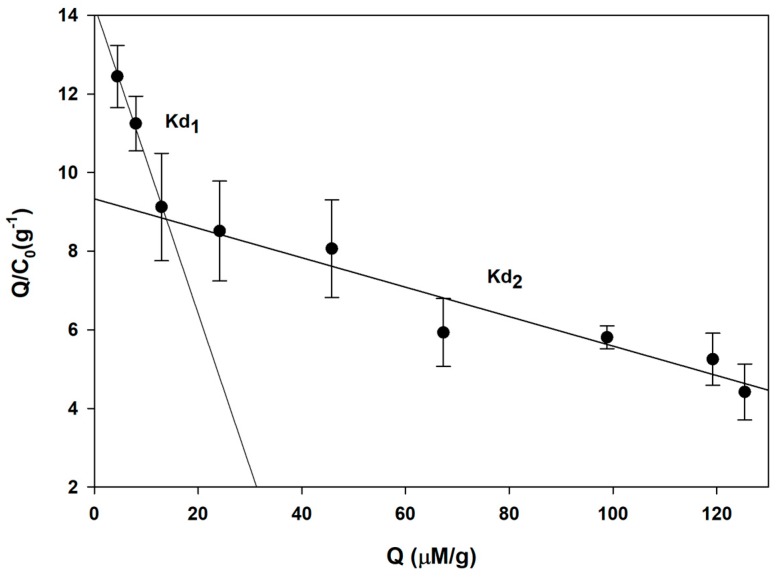
Scatchard analysis of MIPs.

**Figure 6 polymers-11-00570-f006:**
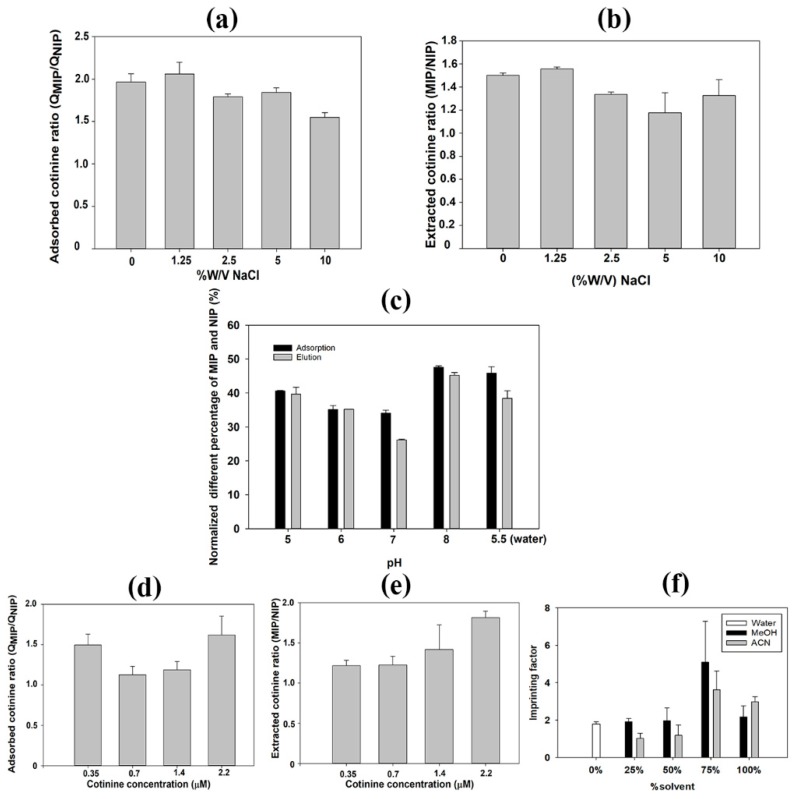
Effects of NaCl, pH, sample matrix, and solvent on cotinine adsorption and extraction. (**a**) Effect of NaCl on adsorbed cotinine ratio between MIPs and NIPs. (**b**) Effect of NaCl on peak area ratio of cotinine extraction. (**c**) Effect of pH on normalized different percentage of MIPs and NIPs. (**d**) Effect of sample matrix on adsorbed cotinine ratio between MIPs and NIPs. (**e**) Effect of sample matrix on peak area ratio of cotinine extraction. (**f**) Effect of MeOH, ACN, and water on cotinine adsorption on MIP/NIP paper-based scaffolds.

**Figure 7 polymers-11-00570-f007:**
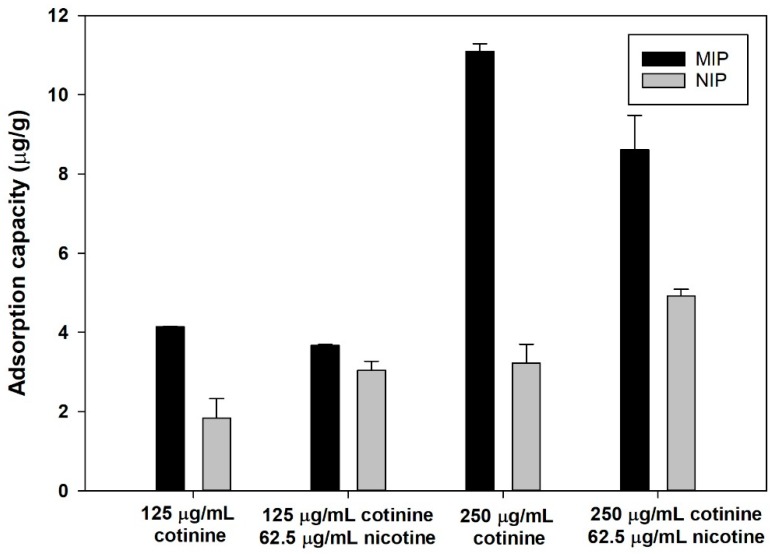
Interference study on adsorption capacity of MIP and NIP.

**Figure 8 polymers-11-00570-f008:**
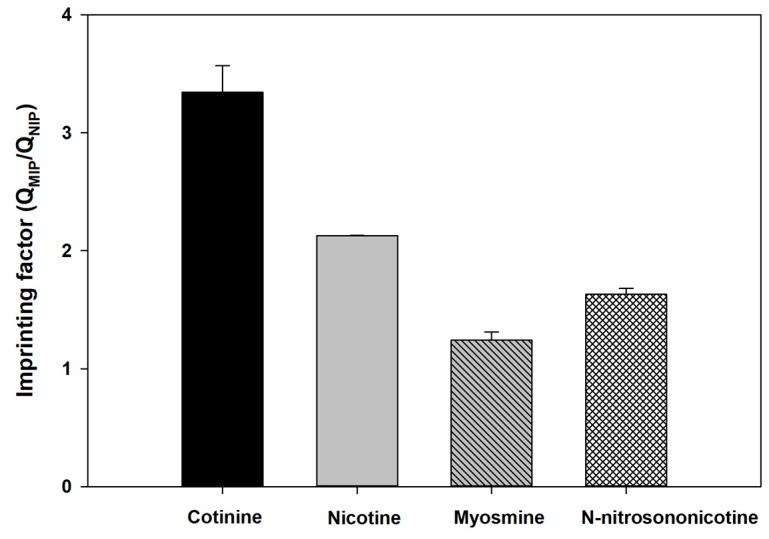
Selectivity of synthesized polymers. Black and gray bars represent imprinting factors for cotinine and nicotine, respectively.

**Figure 9 polymers-11-00570-f009:**
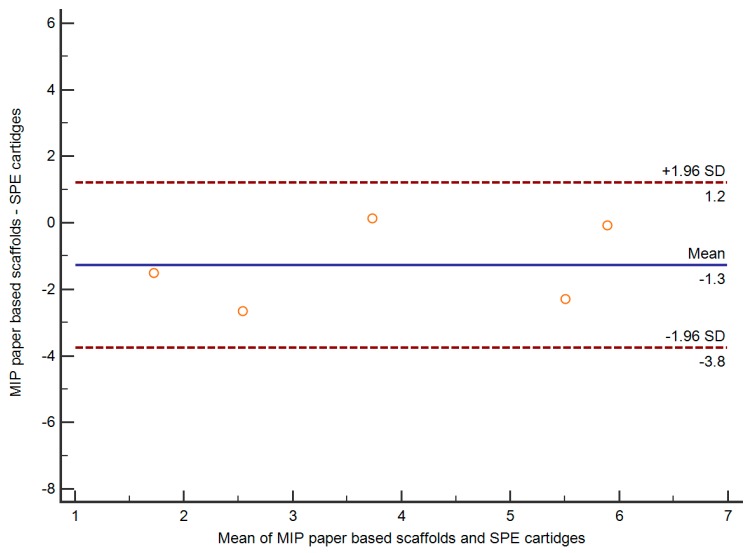
Comparison of results obtained from commercial solid-phase extraction (SPE) cartridges and proposed MIP scaffold.

**Table 1 polymers-11-00570-t001:** Recovery of MIP and NIP paper-based scaffold in urine matrix and water with the presence of nicotine at ratios of 2:1 and 1:1.

	Urine Matrix		Water	
	Cotinine: nicotine (2:1)	Cotinine: nicotine (1:1)	Cotinine: nicotine (2:1)	Cotinine: nicotine (1:1)
**MIP**	88 ± 0.77%	68 ± 0.85%	71 ± 1.6%	70 ± 1.0%
**NIP**	72 ± 2.6%	77 + 3.0%	41 ± 4.0%	50 ± 4.8%

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
