# Peer review of "Exploring Matrix Effects on Binding Properties and Characterization of Cotinine Molecularly Imprinted Polymer on Paper-Based Scaffold"

_polymers, 2019, doi:10.3390/polym11030570_

Round 1

Reviewer 1 Report

The article reports the fabrication of a molecular imprinted polymer as solid phase extraction media for the analysis of cotinine. MIPs were fabricated in bulk and crushed; particles were mixed in agarose gel and supported by stacks of glass fiber filters. 

Neither the MIPs nor the supports are very innovative, however, the application is interesting and the work has merit. 

There are however some concerns that the authors should address: 

-how many cycles of wash and rinse were conducted for the extraction of the template molecule? what was the detection limit of the HPLC technique that was used to monitor template in the wash solutions? Did the authors attempt to perform a mass balance for the target?

- what was the rationale for the concentration of MIP particles that were mixed with the agarose gel in the preparation of MIP paper-based scaffolds? Did the author experiment with other ratios? IF not,  could the final scaffold be further optimized ?

-The authors used cotinine standard solution between 62.5 and 500 microgram /mL. Why were those concentrations used? what is are the expected (or relevant ) levels of cotinine in urine for the intended application? 

- A similar question can be asked for the intereference compounds. How was the ratio of cotinine to interference selected? The authors should select relevant ratios for the proposed application and justify the selected values. 

-The pH effect need to be better explained. The target pKa is 4.5 and the polymer pka~4.8, while the pH range investigated is 5-8, above both pKas.  Differences observed in the 7-8 range cannot be explained solely based on pKa values.

- why is the absorption capacity higher when suspended in agarose gel?

- the specific adsorption on the paper and Nips is not negligible. Recoveris in water and urine matrix for the cotinine and in the absence and presence of interferences should be reported to assess the value of the technique.

-Figure 7, what are the units of the adsorption capacity? 

-Figure S2 needs error bars. 

- there are several grammatical errors and typos throughout the manuscripts. The authors should carefully review the full manuscript and correct those errors  

Author Response

Dear Reviewer,

We thank the reviewer for your careful evaluation of our manuscript and useful remarks, which we found to be thoughtful and appropriate, and feel that attention paid to their concerns will substantially improve our paper. In this revision, the English structure and grammar of the manuscript has been thoroughly reviewed and edited by MDPI Author Services and the highlights were included in the manuscript. We believe that the quality of our manuscript has improved after this revision and are looking forward to further processing of this paper. 

A point-by-point response to the comments made by the reviewer is enclosed.

Reviewer 2 Report

The authors fabricated novel molecularly imprinted polymers (MIPs) on the pape-based scaffold for cotinine adsorption. The conditions optimizing of the cotinine adsorption and extraction process and the interferences and selectivity testing were all investigated. However, several issues have to be solved before its possible acceptance.

1. Fiberglass paper is crucial in this study for cotinine adsorption which used as a paper scaffold. The advantages of fiberglass paper need to be discussed more in details. In addition, for obvious visual perception, the physical photos of glass fiber membrane and MIP paper-based scaffold should be provided.

2. In Figure 3, the ATRFTIR graphs of MIP and NIP should be more complete with exact analysis.

3. The applicability for the MIP paper-based scaffold are not mentioned in the manuscript. Please add experiment to prove applicability of the method.

4. The method performance comparison should be given.

5. There are short of recent research in the reference list. More very recent related publications can be referenced to strengthen the research background or used for improving the manuscript, just for some examples:

Reviews;

Molecular imprinting: perspectives and applications, CHEMICAL SOCIETY REVIEWS 2016 Volume: 45, Issue: 8, Pages: 2137-2211; 

Strategies of molecular imprinting-based fluorescence sensors for chemical and biological analysis, By: Yang, Qian; et al. BIOSENSORS & BIOELECTRONICS 2018 Volume: 112, Pages: 54-71;

Papers:

Rotational Paper-Based Microfluidic-Chip Device for Multiplexed and Simultaneous Fluorescence ……, By: Qi, Ji; et al. ANALYTICAL CHEMISTRY 2018 Volume: 90   Issue: 20   Pages: 11827-11834;

Hydrophilic Multitemplate Molecularly Imprinted Biopolymers Based on a Green Synthesis Strategy for Determination of B-Family Vitamins, By: Ostovan, Abbas; et al. ACS APPLIED MATERIALS & INTERFACES 2018 Volume: 10   Issue: 4, Pages: 4140-4150;

Quantum Dot-Based Molecularly Imprinted Polymers on Three-Dimensional Origami Paper Microfluidic ……, By: Li, Bowei; et al. ACS SENSORS 2017 Volume: 2, Issue: 2, Pages: 243-250;

Negative electrospray ionization ion mobility spectrometry combined with paper-based molecular imprinted polymer disks: ……, By: Zarejousheghani, Mashaalah; et al. TALANTA 2018 Volume: 190   Pages: 47-54;

A molecularly imprinted polymers/carbon dots-grafted paper sensor for 3-monochloropropane-1,2-diol determination, By: Fang, Min; et al. FOOD CHEMISTRY 2019 Volume: 274   Pages: 156-161;

Molecularly imprinted polymer-coated paper as a substrate for highly sensitive analysis using paper spray mass spectrometry: ……., By: Mendes, Thais P. P.; et al. ANALYTICAL METHODS 2017 Volume: 9   Issue: 43   Pages: 6117-6123

Author Response

(The authors gave the same response as above.)
